Animal trait variation at the within-individual level: erythrocyte size variation and malaria infection in a tropical lizard

Cruz Virnaliz virnaliz.cruz@weecology.org virnaliz.cruzher@ufl.edu 1 2
Cruz-Pantoja Omar 3
Tremblay Raymond 4
Acevedo Miguel 1
1 School of Natural Resources & Environment, University of Florida , Gainesville , FL , United States of America
2 Department of Environmental Science, Universidad de Puerto Rico , Rio Pidras , Puerto Rico , United States of America
3 Department of Computer Science, Universidad de Puerto Rico, Recinto de Rio Pidras , San Juan , Puerto Rico , United States of America
4 Department of Biology, Universidad de Puerto Rico , Humacao , Puerto Rico , United States of America
Sorci Gabriele
Electronic publication date: 2022 Feb 23
Publication date: 2022
Volume: 10
Electronic Location ID: e12761
Received 2019 Jul 15; Accepted 2021 Dec 16
Copyright: ©2022 Cruz et al.
Copyright year: 2022
Copyright holder: Cruz et al.
License: This is an open access article distributed under the terms of the Creative Commons Attribution License, which permits unrestricted use, distribution, reproduction and adaptation in any medium and for any purpose provided that it is properly attributed. For attribution, the original author(s), title, publication source (PeerJ) and either DOI or URL of the article must be cited.
License URL: https://creativecommons.org/licenses/by/4.0/

Keywords: Reptile malaria, Plasmodium, Reptile hematology, Anolis gundlachi, Natural-enemy interactions

Funding: The National Science Foundation Graduate Research Fellowship No. DGE-1842473 The University of Puerto Rico Fondo Institucional para la Investigación (FIPI) from the University of Puerto Rico The Puerto Rico Science and Technology Trust Virnaliz Cruz is supported by the National Science Foundation Graduate Research Fellowship under Grant (No. DGE-1842473). The original data set was collected under Miguel Acevedo, and supported by the University of Puerto Rico Fondo Institucional para la Investigación (FIPI) from the University of Puerto Rico and a grant from the Puerto Rico Science and Technology Trust. The funders had no role in study design, data collection and analysis, decision to publish, or preparation of the manuscript.

==============================
High levels of within-individual variation (WIV) in reiterative components in plants such as leaves, flowers, and fruits have been shown to increase individual fitness by multiple mechanisms including mediating interactions with natural enemies. This relationship between WIV and fitness has been studied almost exclusively in plant systems. While animals do not exhibit conspicuous reiterative components, they have traits that can vary at the individual level such as erythrocyte size. It is currently unknown if WIV in animals can influence individual fitness by mediating the outcome of interactions with natural enemies as it has been shown in plants. To address this issue, we tested for a relationship between WIV in erythrocyte size, hemoparasite infection status, and body condition (a proxy for fitness) in a Caribbean anole lizard. We quantified the coefficient of variation of adult erythrocytes size in $n = 95$ infected and $n = 107$ non-infected lizards. We found higher degrees of erythrocyte size variation in infected lizards than in non-infected individuals. However, we found no significant relationship between infection status or erythrocyte size variation, and lizard body condition. These results suggest that higher WIV in erythrocyte size in infected lizards is not necessarily adaptive but likely a consequence of the host response to infection. Many hemoparasites destroy their host cells as part of their life cycle. To compensate, the host lizard may respond by increasing production of erythrocytes resulting in higher WIV. Our results emphasize the need to better understand the role of within-animal variation as a neglected driver or consequence of ecological and evolutionary interactions.

Introduction

Phenotypic traits vary among individuals of the same species. For example, the amount of nectar production (Real & Rathcke, 1991), the density of mycorrhizal fungi on plant roots (Stahl, Christensen & Williams, 1990), and the lethality of venom in animals (Castro et al., 2013) vary among individuals. For centuries, this among-individual variation has been recognized as the building block of natural selection. In contrast, we know little about the eco-evolutionary consequences of within-individual variation (WIV). Until recently, WIV was deemed as a mere statistical nuisance that could be rectified with appropriate experimental design practices and statistical techniques (Herrera, 2009). However, recent studies emphasize the role of WIV in phenotypic traits of plants as a mediator of individual fitness (Austen, Forrest & Weis, 2015; Shimada et al., 2015; Herrera, Medrano & Bazaga, 2015).

Plant fitness has been shown to improve with increasing WIV by mediating ecological interactions with co-evolved symbionts or natural enemies. For example, in the presence of higher degrees of within-plant variation in nectar volume pollinators may limit the number of flowers they visit to avoid the risk of not encountering nectar in the next visited flower (Shafir, 2000). This risk aversion might improve plant fitness by reducing geitonogamy, a type of self-pollination (Biernaskie, Cartar & Hurly, 2002). Similarly, WIV is hypothesized to grant plants a mosaic of protection against natural enemies such as insect herbivores (Whitham, 1983). While insects often have short generations that promote fast adaptation, WIV limits their ability to adapt simultaneously to the spatially complex mosaic of reiterative components (Whitham & Slobodchikoff, 1981). For example, when cotton plants are exposed to leaf damage by the moth Spodoptera littoralis, young top leaves produce more secondary metabolites than older leaves or young side shoots. This variation in response to herbivory by young and old leaves results in a spatial mosaic of secondary metabolites that increase plant fitness by disincentivizing leaf consumption (Anderson & Agrell, 2005). Similarly, mountain birch, Betula pubescens ssp. tortuosa, produces a damage-induced mosaic of resistance by varying the number of secondary metabolites in leaves. When Oporinia autumnata moths were reared using leaves that were adjacent to the mechanically damaged leaves, the larvae showed retarded growth (Haukioja & Niemelä, 1979). With increasing supporting evidence from empirical studies, WIV has been proposed as a key mediator of natural enemies interactions resulting in improved fitness in many plant systems.

This hypothesized relationship between WIV, natural enemies, and individual fitness remains unexplored in animal systems. Animals are not necessarily characterized for having external conspicuous reiterative traits like leaves, flowers, or fruits in plants. Still, vertebrates can exhibit within-individual temporal variation in behavioral traits such as movement activity, sexual displays, or vocalizations (Nakayama et al., 2016; Tanner & Bee, 2019; Bee, 2004). Animals could also exhibit WIV in internal components such as cells size. For instance, Price-Jones (1929), in one of the first and few studies on WIV in non-plant systems, found that humans that suffered from pernicious anemia had higher rates of WIV in the diameter of their erythrocytes compared to healthy individuals. Erythrocyte size variability returned to pre-disease levels after individuals were treated for anemia suggesting that WIV resulted from the disease. In this example, the disease mediated WIV in erythrocyte size. Alternatively, variation in erythrocyte traits can mediate the likelihood of disease infection. For instance, the sickle cell gene is responsible for erythrocyte structural and functional changes. Humans that are heterozygous for the sickle cell gene can still become infected with malaria but have a selective advantage (Allison, 1954). The improved fitness of these heterozygous individuals is explained by variation in sickling rates of Plasmodium falciparum infected and non-infected erythrocytes. The sickling rate of non-infected cells is two to eight times lower (Roth et al., 1978). These results emphasize the role of erythrocytes as reiterative components and hypothesize how a mosaic of erythrocyte sizes can mediate the probability of infection or, alternatively, be the result of disease.

Confirmation of an adaptive role of increased WIV protection against parasite infection would require three types of evidence. First, parasite infection should result in a decrease in host fitness. This could be quantified as a decrease in reproductive output, survival, or sub-lethal effects such as a decrease in body condition (Acevedo et al., 2019). Second, there should be evidence of a difference in the level of WIV between infected and non-infected individuals. Third, individuals with higher degrees of WIV will have higher fitness in comparison with those with lower WIV. Alternatively, if there is evidence of a relationship between infection and WIV without evidence of a relationship with host fitness, it could mean that changes in WIV are a proximate consequence of infection with no adaptive consequence.

To address this issue, we tested for a relationship between WIV in erythrocyte size and Plasmodium azurophilum infection in Anolis gundlachi—a lizard host in Puerto Rico. Specifically, we ask: (1) Do Plasmodium infected lizards have lower body condition? (2) Is infection status related to WIV in erythrocyte size? (3) Is body condition better on individuals with higher WIV in erythrocyte size? This host-parasite system is appropriate to test for this relationship for multiple reasons. First, while P. azurophilum is a common parasite of anoles in the Caribbean, in Puerto Rico ∼90% of infections occur on A. gundlachi. This level of specialization and other empirical evidence suggests that this host-parasite system has co-evolved for many years (Schall, 1990; Schall, Pearson & Perkins, 2000). Second, the host lives in high abundances (2,000 ind/ha; Reagan & Waide, 1996) allowing for the large sample sizes required to properly quantify WIV. Third, the population that we studied is protected from human intervention, which helps control for anthropogenetic pressures.

Methods

Study site and species

To answer these questions our data consists of a subset of individuals from an ongoing long-term study on the dynamics of the Anolis–Plasmodium host–parasite system at El Verde Field Station, in the Luquillo Experimental Forest in Puerto Rico (18.3213° N, 65.8194° W, WGS 84; 357.9 m elev) (Schall, Pearson & Perkins, 2000; Otero et al., 2019). The surveys took place during the summer season of 2015 and the winter season of 2016. The Institutional Animal Care and Use Committee protocols from the University of Puerto Rico provided the permits for the handling of lizards, IACUC (01005-01-09-2015).

Out of the seven species of anoles inhabiting El Verde, we selected Anolis gundlachi as our study species because it is the member of the anole community with the highest prevalence of Plasmodium parasite infection in the site. Between 2015 and 2017 the probability of infections was 0.10–0.19 for males, and 0.06–0.12 for females (Otero et al., 2019). This anole is a medium-sized montane lizard, part of the trunk-ground ecomorph, with a snout-vent length (SVL) range of 42–72 mm (Reagan & Waide, 1996). This species can be infected by three Plasmodium species: P. azurophilum, P. floridense, and P. leucocytica. We analyzed infections by P. azurophilum because this is the most common blood parasite at the site, accounting for 60–80% of infections (Schall, Pearson & Perkins, 2000; Otero et al., 2019). Also, P. azurophilum is the only member of this Plasmodium community known to destroy lizard erythrocytes (Schall, 1996). Co-infections are infrequent. Still, we restricted our analysis to single infected individuals to ensure our results were not biased by potential interacting effects of multiple blood parasites (see below).

We captured anoles by hand or using the lasso technique while the lizards were perching on tree trunks, branches, or on the ground. The lizards were briefly kept in individual bags and moved to the laboratory where they were processed to identify sex, measure the snout-to-vent length (SVL) and weight, and obtain a blood sample via toe clip (Schall & Vogt, 1993). We used this blood sampling method because it provides enough blood to accurately diagnose infection by microscopy and PCR (see below). Also, toe clipping provides a permanent mark that prevents sampling of an individual more than once. Elevated levels of corticosterone have been shown to alter the hematopoietic cell composition and immune response in other animals; however, this method has less impact on corticosterone levels than implanted Passive Integrated Transponder (PIT) tags (Langkilde & Shine, 2006). All lizards were released back to the general area in which they were found within 24hrs.

Infection diagnostics

We diagnosed Plasmodium infection using microscopy and verifying through polymerase chain reaction (PCR). Combining both methods assures the most accurate classification of infection status. For microscopy diagnosis, we made thin-blood smears on glass slides. The thin blood smears were fixed with 100% methanol for a minute and stained with Giemsa at a pH of 7.5 for 50 min. We examined the blood smears at 1000×magnification using a Nikon Eclipse E2000 light microscope for 6–10 min (Otero et al., 2019). We classified the Plasmodium species detected by morphological traits (Telford, 2009). We considered a lizard infected if we detected through microscopy at least one cell invaded by P. azurophilum, and if PCR confirmed infection by positively amplifying Plasmodium. Similarly, we considered a lizard non-infected if we could not visually detect an invaded cell using microscopy and the PCR did not amplify Plasmodium. To ensure the most accurate classification of infection status, if there was a mismatch between the slide blood smear and the PCR result the individual was removed from the analysis.

Drops of lizard blood were spotted on Whatman filter paper and stored at −20 °C in sealed bags with silica gel for further PCR testing. DNA was extracted from blood-spot filters using the DNeasy Extraction kit with some modifications to the manufacturer’s instructions. To extract the blood from the filter paper, 180 µL of Buffer ATL and 20 µL Protease K was added. The samples were mixed and incubated at 56 °C overnight with shaking. The next day we added 200 µL of Buffer AL to the sample, mixed by thoroughly shaking for 15 s, and incubated at 56 °C for 15 min. Then, 200 µL ethanol was added to each sample. To elute DNA, we added 50 µL to 75 µL Buffer AE to each sample and incubated it for 5 min at room temperature. The extracted DNA(eluate) was stored at −20 °C. A partial fragment of the mitochondrial cytochrome b gene was amplified using nested PCR (nPCR). The nPCR was done with Taq PCR Master Mix kit (Qiagen cat 201445) under the conditions described by Perkins & Schall (2002) with minor modifications. Briefly, an outer reaction using primers DW2 F 5′- TAA TGC CTA GAC GTA TTC CTG ATT ATC CAG - 3′ DW4 R 5′- TGT TTG CTT GGG AGC TGT AAT CAT AAT GTG - 3′ and 2 µL of genomic DNA was subject to thermal cycling program included an initial denaturation step at 94 °C for 4 min followed by 35 cycles of 94 °C for 20 s, 60 °C for 30 s, and 72 °C for 1.5 min. The nested reactions were done with 1 µL of the previous PCR product and primers DW1 5′-TCA ACA ATG ACT TTA TTT GG-3′ and DW6 5′-GGG AGC TGT AAT CAT AAT GTG-3′ under initial denaturation step at 94 °C for 1 min followed by 40 cycles of 94 °C for 20 s, 50 °C for 20 s, and 72 °C for 1,5 min and then 72 °C for 7 min. Positive and negative reaction controls were included. Amplification products were resolved in a 2% agarose gel stained with gelStar™ (Lonza, cat # 50535) and visualized under ultraviolet light. The primers only amplified the malarial parasite DNA. PCR complemented diagnosis by helping avoid false negatives resulting from individuals with low parasitemia that may be missed by the slides.

Erythrocyte image capturing and measurement

We photographed 10 microscope optical fields for each lizard’s thin blood smear (Fig. 1). Optical fields were selected at random. Fields in which less than 10 mature erythrocytes were measurable were discarded. We measured only mature erythrocytes in each picture and disregarded other hematocytes because we wanted to estimate variation in erythrocyte size independent of stage. We took the pictures at 40x power using a Nikon DS-Fi2 camera head and a Nikon DS-U3 microscope camera controller. This setting provided high-quality pictures that allowed precise measurements of the erythrocyte area. Since measuring erythrocyte size in clumps of cells is challenging and may result in inaccurate measures, we considered cells to be measurable if there was at least 0.070 µm of space between cells, and if they were not damaged during the smear making process. We also discriminated between infected and non-infected erythrocytes. We discarded 19 Plasmodium-invaded cells (out of a total of 50,241 measured erythrocytes) from the analysis because erythrocyte sickling rate has been observed to differ between Plasmodium stages (Roth et al., 1978). By restricting our analysis on mature erythrocytes that were not invaded by the parasite we segregated the processes allowing us to assess the global effects of Plasmodium on WIV independent of local effects.

Figure 1 To quantify the within-individual variation in lizard erythrocyte size, we photographed thin blood smear optical fields of P. azurophilum infected and non-infected A. gundlachi individuals.

The figure shows an example of photographed optical field of a thin blood smear sample at a 40×magnification of A. gundlachi individual. The numbers represent erythrocyte areas (µm2). We measured mature erythrocytes and calculated the coefficient of variation of cell size per lizard.

We measured the area of each erythrocyte in each optical field with the image processing software Image J (Schneider, Rasband & Eliceiri, 2012). We set the scale to 50.0 µm and the image type to 8-bit grayscale to allow segmentation by pixel color intensity, also known as image thresholding. The image threshold was manually adjusted for each image due to differences in the sharpness, quality of staining, and brightness of images. We selected the optimum threshold that allowed us to minimize measurement error by decreasing the amount of whitespace or non-erythrocyte area inside the measurement polygon. We stored measurements of erythrocyte per picture automatically on individual spreadsheets with their unique ID as the file name. We also took lower resolution pictures of the selected erythrocytes per picture which had the order in which erythrocytes were measured. This allowed us to keep a permanent record of which measurement belonged to which erythrocyte for verification purposes while maintaining our original pictures intact to ensure study reproducibility.

Data analyses

We conducted a prospective power analysis to estimate an appropriate sample size to test our hypotheses with statistical power >0.8. We analyzed pilot data (n = 21) to estimate the variance and mean coefficient of variation of the population. To estimate the effect size, we used Cohen’s d defined as, d=x ¯inf−x ¯ninfSDpooled, where x ¯inf=12.59 (infected CV) and x ¯ninf =12.02 (non-infected CV). We set a significance level of α = 0.05 and a power of 0.8 as it is conventionally done in studies of ecology and evolution.

We quantified within-individual erythrocyte size variation of A. gundlachi using the coefficient of variation (CV), such that:

CVi=sixi ¯×100,

where x ¯i represents the sample mean and si the sample standard deviation of the size of mature erythrocytes of lizard i. The coefficient of variation (CV) is a measure of relative variation allowing comparison among lizards while controlling for mean erythrocyte size (Donnelly & Kramer, 1999). This measure is commonly used in similar studies to quantify within-individual variation (Biernaskie, Cartar & Hurly, 2002; Herrera, Medrano & Bazaga, 2015; Langkilde & Shine, 2006).

We modeled lizard body condition index (BCI) as a function of infection status, and sex and season as controlling variables using a multiple linear regression. Although reproduction data would be ideal to test for the negative consequences of infection on individual fitness, collecting breeding data is challenging and often infeasible in many natural systems including anoles (Jakob, Marshall & Uetz, 1996). Here we use BCI as a proxy for fitness as the next best alternative. We calculated BCI as the residuals of the linear regression of log10 weight (g) explained by log10 lizard SVL (mm) (Cox & Calsbeek, 2015). Positive residuals suggest a better body condition relative to the population average (Schall, Pearson & Perkins, 2000). Male A.gundlachi BCI has been shown to be higher than females and BCI also responds to seasonal variation in resource availability (Schall, Pearson & Perkins, 2000). Previous studies validate the use of BCI to represent the energetic state of animals (Ardia, 2005; Schulte-Hostedde et al., 2005). Still, a study found that BCI is weakly related to survival in Anolis sagrei (Cox & Calsbeek, 2015) and a meta-analysis found small, negative average effect sizes between BCI and parasite infection that was stronger in laboratory settings (Sánchez et al., 2018). In an experimental setting, mass/SVL was found to be the most accurate estimator for the mass of chemically extracted fat for a reptile among five compared estimators (Sion, Watson & Bouskila, 2021). Therefore, while BCI is the best approximation to fitness given our data constraints we are careful interpreting the results given its mixed support as a proxy for fitness (Wilder, Raubenheimer & Simpson, 2016).

To test for a relationship between P. azurophilum infection and WIV in erythrocyte size, we modeled infection status as a function of CV. Previous studies show that larger males in the summer season have a higher probability of getting infected (Otero et al., 2019). Therefore, we included sex, season, and body size(SVL) as controlling covaraites. We modeled this relationship using a generalized linear model with a Bernoulli distribution and logit link function.

To test if individuals with higher WIV have improved fitness, we modeled BCI as a function of CV in erythrocyte size with sex and season as controlling covaraites using a multiple linear regression. These controlling covariates were included because previous studies in this system found that body condition is higher in males and during the summer season where food resources for lizards are more abundant (Schall, Pearson & Perkins, 2000).

Results

In the power analysis to determine appropriate sample size, infected individuals’ erythrocyte size was more variable (CV = 12.59 ± 1.74 SD) than non-infected individual erythrocyte size (CV of 12.02 ± 1.27 SD). With an estimated effect size of d = 0.38, the power analysis suggested an optimal sample size of N = 112 individuals in each category (infected and non-infected). This sample size was a conservative estimate that falls under the small-medium effect size category (0.20 standard deviation difference is considered small, and 0.50 standard deviation difference is considered medium) (Cohen, 1992). Our final data set allowed us to analyze a total Ninfected = 95 and Nnoninfected = 107 individuals after discarding individuals with co-infections or poorly stained slides. Therefore, our overall sample size was appropriate to attain appropriate power.

A linear model predicted no significant relationship between infection status and BCI (βinfected = 0.015 ± 0.01SE, P = 0.14; Fig. 2). Similarly, this model predicted no significant effect of season on BCI (βwinter =  − 0.02  ±  0.01SE,  P = 0.05). Still, the model predicted that males would have higher body condition than females (βmales = 0.03 ± 0.01,  P = 0.01).

Figure 2 The predicted partial relationships between infection status and body condition index (BCI) for female and male A. gundlachi during the (A) 2015 summer (B) and 2016 winter seasons.

Grey error bars represent 95% CI and black error bars represent one standard error.

We found that the CV of erythrocyte size is a strong predictor of infection status (z = 2.92,  P = 0.003, Fig. 3). The binomial model predicted that the probability of being infected increases 1.43 (1.13–1.83 95% CI) times with a unit increase in CV. The model also predicted that the probability of being infected is 0.63 (0.34–1.16 95% CI) times in the winter compared to the summer, but this relationship is uncertain (z =  − 1.48,  P = 0.14). Similarly, the model predicted that the probability of being infected is 0.79 (0.33–1.9 95%CI) times in males when compared to females, but this relationship was also uncertain (z =  − 0.52,  P = 0.60). The probability of infection is predicted to increase 1.08 (1.03–1.12 95% CI) times with a mm increase in SVL (z = 3.3,  P < 0.001; Fig. 3).

Figure 3 The model predicts that the probability of infection increases with increasing WIV quantified here as CV in erythrocyte size.

The figure shows partial predictions for male and female A. gundlachi during the (A) 2015 summer season and (B) 2016 winter season. The size of the points represents relative SVL and their color infection status.

While we found CV of erythrocyte size to be a key predictor for the probability of infection, CV was not a significant predictor of BCI (β =  − 0.001 ± 0.004SE,  P = 0.86; Fig. 4). Similar to the analysis presented above on BCI predicted by infection status, season was not a significant predictor of BCI (βwinter =  − 0.02 ± 0.01SE,  P = 0.08), but males had better body condition than females (βmales = 0.03 ± 0.01SE,  P = 0.006).

Figure 4 The model predicts no significant relationship between CV of erythrocyte size and BCI.

The figure shows partial predictions for female and male A. gundlachi during the (A) 2015 summer season and (B) 2016 winter season. The size of the points represents relative SVL.

Discussion

Studies in plant systems show that WIV increases individual fitness through multiple mechanisms including mediating the outcome of interactions with natural enemies. Testing this hypothesis in animal systems requires studying reiterative structures that can be exposed to selective processes. Here we studied the relationship between WIV in erythrocyte size, hemoparasite infection, and body condition in the Caribbean lizard Anolis gundlachi. Our results show that infected individuals had a higher WIV in erythrocyte size than non-infected individuals, but WIV was not related to the host body condition. This result suggests that the higher WIV in infected may not necessarily be adaptive but likely a consequence of infection.

There are three potential explanations for the lack of variation in BCI due to infection status or WIV in erythrocyte size. First, virulence—the negative consequences of infection for host fitness—may be low or undetectable in this system. Virulence varies widely in lizard malaria systems. For example, P. mexicanum infection on Sceloporus occidentalis in California, USA and P. agamae infection on Agama agama in Sierra Leone, West Africa results in lower hemoglobin levels and a decreased in physiological characteristics such as oxygen consumption and stamina (Schall, 1990). In contrast, there are few indications of virulence to Caribbean hosts. For instance, Anolis sabanus infected by P. azurophilum had no difference in body temperature, broken tails, habitat use, or intraspecific interactions with non-infected individuals (Schall & Staats, 2002). Similarly, other studies have found little evidence of decreases in body condition due to infection in Caribbean lizards (Pearson, 2000; Otero et al., 2019). This variation in virulence is also present in Plasmodium infection to avian hosts. For example, Plasmodium infection is linked to a dramatic decrease in Hawaiian honeycreeper population size (Atkinson et al., 1995). Still, chronic infections show no important negative consequences for American songbirds (Matthews et al., 2016). Second, our results show that infected individuals have, on average, better body condition, but the uncertainty is too large to make this pattern statistically significant. This result could also mean that A. gundlachi is slightly tolerant of P. azurophilum infection, but the effect size is too small to be easily detected. Tolerance to parasite infection is the result of individuals investing in physiological mechanisms that limit fitness costs of infection without limiting parasite reproduction which has been proposed for other lizard malaria systems (Rausher, 2001; Bonneaud et al., 2017). Third, BCI—our proxy for fitness—may be weakly related to fitness in A. gundlachi. A recent meta-analysis shows large variability in how body condition indices relate to fitness in many host-parasite systems (Sánchez et al., 2018). Still, the lack of evidence for Plasmodium virulence to other Caribbean lizards is consistent. The study of Plasmodium spp. virulence to Anolis sabanus in Saba, an island geographically close to Puerto Rico, found no evidence of virulence looking at a wide array of variables which supports our results (Schall & Staats, 2002).

Overall, higher degrees of WIV in infected individuals might be explained by multiple types of interactions between A. gundlachi and Plasmodium. In humans, Plasmodium infection induces structural and functional changes in erythrocyte traits such as loss of discoid shape, increased membrane rigidity, elevated permeability, reduced deformability, and increased adhesiveness (Cooke, Mohandas & Coppel, 2001). The higher WIV erythrocyte size variation in infected A.gundlachi may also be explained by induced erythropoiesis—erythrocyte production. Plasmodium infection results in the global destruction of erythrocytes and consequently the lizard may respond increasing the production of immature erythrocytes in the hemopoietic bone marrow (Zapata, Leceta & Villena, 1981; Zivot et al., 2018). This process may lead to higher size variability in erythrocyte size in infected lizards compared to their non-infected counterparts. Higher numbers of immature erythrocytes have been found at weak and moderate degrees of parasitemia in a variety of Plasmodium lizard hosts including Sceloporus occidentalis, Agama agama and Anolis gingivinus (Schall, 1990; Schall, 1992). Still, we restricted our analysis to adult erythrocytes. Therefore, any observed effect of increased erythropoiesis in our system is not due to an increase in immature cells, but the residual effect on adult cells. While increased erythropoiesis is a parsimonious explanation for the increased WIV in infected lizards, WIV in animals could be the result of other complex interacting processes such as organ-level developmental plasticity, diet, reproductive state, metabolic rate, or genetic relatedness.

The study of the drivers and consequences of WIV is an emerging sub field in ecology and evolution (Herrera, Medrano & Bazaga, 2015). Most of the evidence supporting the role of WIV comes from plant systems (Herrera, 2009). The apparent lack of obvious reiterative structures in animal systems limits our ability to generalize this theoretical framework. This study aims to contribute to the expansion of the study of WIV in animal systems by testing the role of erythrocytes as a reiterative internal structure. While we found a clear relationship between WIV in erythrocyte size and parasite infection status, our results showed no clear evidence that this WIV has an adaptive role. Likely the increase in WIV is a consequence of the host’s response to infection. Still, the potential adaptive role of WIV in animal systems remains a wide and unexplored area of research. The mere presence of WIV on any system can have profound implications on sampling techniques and experimental designs used to study it. We hope our study encourages additional studies in other systems.

Conclusions

This is one of the few studies aimed to understand the drivers and consequences of within-individual variation in animal systems. We tested for the role of variation in erythrocyte size as an internal reiterative component potentially mediating interactions with natural enemies in a Caribbean lizard-malaria system. While we found a clear relationship between WIV and infection status, the lack of evidence of changes in body condition suggests that erythrocyte size variation is likely a consequence of infection and not necessarily adaptive.

Supplemental Information

Supplemental Information 1 WIV in a Tropical Lizard: Code and Results

Click here for additional data file.

We would like to thank Elizabeth Evans, Dr. Gary Gervais, Wilson E. Lozano, Dr. Luisa Otero, Dr. María E. Pérez, Ana C. Medina, Dr. Ana Trujillo, Dr. Ethan P. White, Dr. Jos Schall, and Dr. Hao Ye for all their invaluable help.

Additional Information and Declarations

Competing Interests

Author Contributions

Animal Ethics

Data Availability

The authors declare there are no competing interests.

Virnaliz Cruz conceived and designed the experiments, performed the experiments, analyzed the data, prepared figures and/or tables, authored or reviewed drafts of the paper, assisted in making program and UI for removal of immature erythrocytes, and approved the final draft.

Omar Cruz-Pantoja analyzed the data, authored or reviewed drafts of the paper, assisted in making program and UI for removal of immature erythrocytes, and approved the final draft.

Raymond Tremblay conceived and designed the experiments, authored or reviewed drafts of the paper, and approved the final draft.

Miguel Acevedo conceived and designed the experiments, analyzed the data, prepared figures and/or tables, authored or reviewed drafts of the paper, and approved the final draft.

The following information was supplied relating to ethical approvals (i.e., approving body and any reference numbers):

This research was conducted under the permits of Institutional Animal Care and Use Committee protocols from the University of Puerto Rico (01005-01-09-2015).

The following information was supplied regarding data availability:

The data is available at Zenodo: Virnaliz Cruz, & Miguel A. Acevedo. (2019). Within-lizard erythrocyte size variation [Data set]. Zenodo. https://doi.org/10.5281/zenodo.5484245.

The code is available in GitHub: https://github.com/garezana/wiv-anolis-plasmodium.

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
