# Peer review of "Animal trait variation at the within-individual level: erythrocyte size variation and malaria infection in a tropical lizard"

_PeerJ, doi:10.7717/peerj.12761_

## Round 0.1 · original submission · Major Revisions

Both referees raised a number of points mostly related to the rationale underlying the study, and I agree with them. Based on my own reading of the manuscript, I think that the conceptual framework of the study should be better explained. The starting point is previous work done in plants. You provide a few examples of adaptive within-individual variation and in this case, the underlying mechanisms are clearly identified. However, the reasons why we may a priori expect that within-individual variation in red blood cell (RBC) size protects towards Plasmodium infection are not clear at all. If any, my prediction would have been that there should be a positive correlation between within-individual variation in RBC size and infection, simply because infection produces the lysis of RBCs and stimulates the production of reticulocytes (which are bigger than mature RBCs). Therefore, to me, a meaningful comparison between infection and within-individual variation in RBC size should control for RBC age (focusing only on mature RBCs). I would also suggest being very cautious about the adaptive nature of a possible correlation between within-individual variation in RBC size and infection. First, if as I suspect, the correlation emerges only because of the size variation of RBCs of different ages, then there is no adaptation going on here. Second, using BCI as a proxy of fitness is not that straightforward and, as suggested by referee 1, you should at least provide evidence showing that infected lizards have lower BCI than non-infected individuals.

Reviewer 1 ·

Basic reporting

The manuscript is well written and is clear and concise on an interesting and clearly understudied topic. The introduction and background provide an important foundation for the reason for the manuscript, and clearly explain why it may be important to study the topic of “within individual variation” in animals. Please see my comment in the “General Comments” on where it may include more background. Overall, the manuscript fits PeerJ’s standards as well as scope. The attached figures, code, and data are high quality, easy to understand, and very clear.

Experimental design

The experimental design is straightforward and easy to understand. This combination means that the results make sense and the paper works well to answer the question they originally asked. The methods are regularly used, generally explained well, and are replicable. Please see “General Comments” for questions about the methodology.

Validity of the findings

The authors asked an interesting and important question, and this is particularly evident in the findings. Any information found using their solid techniques would be both interesting and move the field forward (due to an extreme lack of WIV examples in vertebrates). The underlying data and code are provided, including statistical modelling code. The authors make sure to explain their findings and give several reasons as to why the data are the way they are.

Additional comments

Overall, I thought this was a well written and interesting paper on a topic that is rarely studied, especially outside of humans. While Plasmodium infection is relatively well studied, this manuscript does two important things: looks at how within individual variation may be related to Plasmodium infection and introduces within individual variation into the animal literature. Please see my comments below on how it can be improved and clarified.

1. The statistics are valid and easily reviewed with the provided code. I think the authors need to do a better job at explaining why they did the stats though. This is because there are plausible ways to do very different analyses. For example, the current analysis is based on the hypothesis that within individual variation will predict Plasmodium infection (prediction based off the plant literature). The introduction explains a human example where within individual variation is caused by a disease though, and that could mean you could switch the model such that Plasmodium infection would predict within individual variation. The authors should make the argument stronger for why they are doing the model based on the plant hypothesis, and this may be improved using the literature brought up in the second comment below.
2. The authors should include more examples of within individual variation in animals. This could include examples that is well studied in the literature, such as sickle cell anemia in humans. This particular example should be included because it is both relevant to the topic of within individual variation and Plasmodium infection (i.e. humans with sickle cell trait show high within individual variation of erythrocytes and this lowers Plasmodium infections). Including this will also provide a stronger biological reason for your hypotheses and predictions (i.e. the hypothesis that higher within individual variation would be associated with lower infection).
3. Is Plasmodium infection related to BCI in the lizards in this study? Despite there being debate about this in the literature, this should be included in the manuscript, rather than simply citing Sanchez et al. 2018. This is because the authors are saying high WIV may be related to fitness via BCI. But if infection is not related to BCI at all, why would you test if WIV could be a proxy for fitness/infection? Particularly it would be important to note because on line 200, the authors state that there is a small negative effect size (according to Sanchez et al. 2018), so it would be good to know what it is in this study.
4. There is some confusion about how lizards were assigned to be infected or not infected. Starting on line 119, the manuscript states “We considered a lizard infected if we detected through microscopy at least one cell invaded by P. azurophilum, and PCR positively amplified Plasmodium. We considered a lizard non-infected if we could not visually detect an invaded cell using microscopy and the PCR did not amplified [sic] Plasmodium.” This phrasing makes it sound as if to be identified as positively infected, a lizard had to have Plasmodium visually seen AND PCR had to amplify the parasite DNA (not OR). If it was one or the other, but not both, the lizard would be marked as non-infected. Later, on line 143, the authors state “PCR complemented diagnosis by helping avoid false negatives resulting from individuals with low parasitemia that may be missed by the slides”. It is unclear if the authors meant to say positively infected individuals could be found either through visual inspection or PCR.
5. In the Discussion section, there are several lines that are very similar and can be condensed to be less repetitive and clearer. Lines 255-257 are very similar to 265-267.
6. Small typographic error on line 260 (no space after period), and a small grammatical error on line 122.

Reviewer 2 ·

Basic reporting

This study submitted to PeerJ addresses the hypothesis that a high within-individual variation (WIV) in reiterative components increases the individual fitness by mediating ecological interactions. The novelty of the study is its application to animal systems, a matter so far poorly studied. Authors investigated whether the area of reiterative components (i.e., erythrocytes) can mediate and reduce successful infections by Plasmodium azurophilum in the lizard host Anolis gundlachi.

Experimental design

The hypothesis is based on the idea that tight co-evolutionary histories obligate parasites to be highly specialized and, thus, encountering target components with a high variability reduces the parasite’s capability of infection and increases host's fitness. This is an interesting approach although I find (in page 118 of Sam R. Telford, Jr.’s 2009 book “Hemoparasites of the Reptilia: Color atlas and text”, CRC Press, Taylor and Francis Group, Boca Raton, FL) that Plasmodium azurophilum is not a specific parasite of A. gundlachi, and actually is a generalist known to infect 20 different species of Anolis which probably vary in their erythrocyte area. As it is known that close related lizards vary in their erythrocyte area depending on e.g. environmental conditions. All this said makes me think that this may not be a good model to test your hypothesis.

Besides this issue, I also think that the infection by Plasmodium, and first reproduction cycles, take place first in the liver (already damaging it and producing a negative impact) during the preerythrocytic phase. Thus, although hosts had a high WIV concerning their erythrocyte size I do not find how this could reduce the infection process and the negative impact on the liver. Maybe this could reduce, at some point, during the Plasmodium cycle, the invasion of erythrocytes and subsequent transmission to a new feeding mosquito. However, following the results achieved it neither seems to be the case because, contrary to predictions, WIV was higher in infected individuals.

In addition to these two major issues, I suggest calculating BCI for all the data together and not splitting the calculus for sex and season. You are interested to compare whether individuals are above or below the sample mean, if you split the calculus of BCI by group there is no point in comparing the values of BCI because you are making the subsamples independent (which also makes no sense including sex and season in the analysis, because you are already removing their effect).

Validity of the findings

I am afraid I cannot be more positive. The results concerning erythrocyte size are good but the approach of the study and the hypothesis should be reformulated. I have to mention that I agree with the interpretation of the results: WIV is higher in infected individuals probably due to the higher cell replacement rate.

The results concerning BCI should be recalculated based on my prior comments.

Additional comments

I think that the study could be refocused to tell a different story because you got a lot of data and I know that screening smears and DNA samples is a lot of work. I find very interesting your results suggesting that parasites destroy erythrocytes, which we already know that this happens, and lizards have to compensate the destruction of cells by replacing them (with the added cost). Maybe, refocusing this study, you can test the cost of the replacement of the destroyed cells comparing the ratio of immature erythrocytes against BCI (after you recalculate BCI for the entire sample).

I find Peig and Green (2009, calculation of SMI) in References but not in the text. Probably to be deleted.

---

## Round 0.2 · Minor Revisions

Although the referee agreed that this is an improved version, she/he still provided several suggestions that need to be considered in a further revision. In particular, I urge you to be particularly careful when stating that you assessed lizard fitness.

Reviewer 2 ·

Basic reporting

The authors provide a complete revised version. I appreciate their effort. However, I still have some few comments before I can recommend publication.

Line 31. I would rather write ‘the host may respond by increase production’. I don’t think that the immune system is responsible in any way of producing erythrocytes.
Line 204. You did not really quantified fitness but a proxy of it. I would rather write here ‘We modelled body condition…’, and remove the first part of the phrase, which was already explained in Introduction.
Lines 212-214: This is redundant information ‘Therefor, while we calculate a pooled BCI we added sex and season as controlling variables in the linear model explaining the relationship between infection status and BCI.’
Line 215: The two references provided support the use of body condition indices as proxy to the ‘energetic state of animals’. However, I would prefer to have examples cited on lizards, or at least reptiles.
Line 219: You may prefer to cite something like: ‘Wilder, S. M., Raubenheimer, D., & Simpson, S. J. (2016). Moving beyond body condition indices as an estimate of fitness in ecological and evolutionary studies. Functional Ecology, 30(1), 108-115.’, rather than the message ‘see below’.
Lines 225: Please, remove ‘To test if individuals with higher WIV have improved fitness,’. This information might be better placed in Discussion.
Lines 280-282: an alternative hypothesis is that only lizards in better condition survived the infection. Indeed, the destruction of erythrocytes (and the need for replacement) is not a tolerance mechanism in any way, but rather a response mechanism to the infection demonstrating host vulnerability! I think that the following lines relating tolerance mechanisms should be shortened.
Line 298: This is a misconception. The immune system is not responsible in any way of erythropoiesis.
Line 315: Remove ‘host immune response to’.
Line 324: the authors provide no measure of fitness. Please, rephrase.
Figure 1. Could you provide a similar picture showing some infected erythrocytes?

Experimental design

no comment

Validity of the findings

I recommend revise some of the interpretations provided by the authors.

---

## Round 0.3 · accepted · Accept

The few pending issues have been addressed in this second revision.